# Climate Change and the Herbicide Paradigm: Visiting the Future

**L. H. Ziska**

Environmental Health Science, Mailman School of Public Health, Columbia University, New York, NY 10032, USA; lhz2103@cumc.columbia.edu

**Abstract:** Weeds are recognized globally as a major constraint to crop production and food security. In recent decades, that constraint has been minimized through the extensive use of herbicides in conjunction with genetically modified resistant crops. However, as is becoming evident, such a stratagem is resulting in evolutionary selection for widespread herbicide resistance and the need for a reformation of current practices regarding weed management. Whereas such a need is recognized within the traditional auspices of weed science, it is also imperative to include emerging evidence that rising levels of carbon dioxide ($CO_2$) and climatic shifts will impose additional selection pressures that will, in turn, affect herbicide efficacy. The goal of the current perspective is to provide historical context of herbicide use, outline the biological basis for $CO_2$/climate impacts on weed biology, and address the need to integrate this information to provide a long-term sustainable paradigm for weed management.

**Keywords:** climate; $CO_2$; evolution; herbicides; paradigm; resistance

---

## 1. Introduction

At present, it is necessary to provide a sufficient and continuous food supply for a global population approaching 8 billion. By 2050, the global population is expected to increase to ~10 billion [1].

Providing food security for an increasing global population remains a fundamental challenge for humanity, and meeting that challenge will necessitate overcoming a number of obstacles, including shifts in diet, diversion of crops to biofuels, loss of agricultural land, and anthropogenic climate change.

Overall, climatic uncertainty and potential extremes will pose inherent environmental risks in regard to agricultural production and sustainability [2,3]. Such risks, including water availability, excessive or irregular temperatures, and extreme climate events, have been the subject of considerable research [4–7].

Overcoming climate induced physical barriers to maintain and sustain crop production is of obvious importance [8]. Yet, it is also worth emphasizing that rising levels of carbon dioxide and climate change will impact biological constraints to crop production including weeds, insects, and diseases [9]. Such biological impediments represent a significant constraint to global food production. For example, recent estimates [10] for pest and pathogen losses average 21.5% for wheat, 30% for rice, 17% for potato, and 22.5% for maize globally. A number of investigations have indicated that the production risk posed by biological threats is likely to be exacerbated with rising $CO_2$ and climatic change (insects, [11]; diseases, [12]; weeds, [13]).

The exacerbation of pest constraints on production by climatic change necessitates a closer examination of future pest management efforts. Specifically, are current management paradigms sufficient to minimize pest impacts in the context of a changing climate?

## 2. Weed Management

Among biological impacts, weeds represent a major limitation to crop production (Figure 1). Weeds share a similar trophic level with crop plants and can compete for similar resources with disproportionate reductions in crop yields relative to other pests. In addition, they may also compromise crop quality directly through contamination or act as a vector for additional pest infestation (i.e., viruses) [14,15]. Yet, it is also clear that such potential losses do not occur due to weed control efforts. Indeed, the largest relative reduction in pest pressures due to management is associated with weed control (Figure 1).

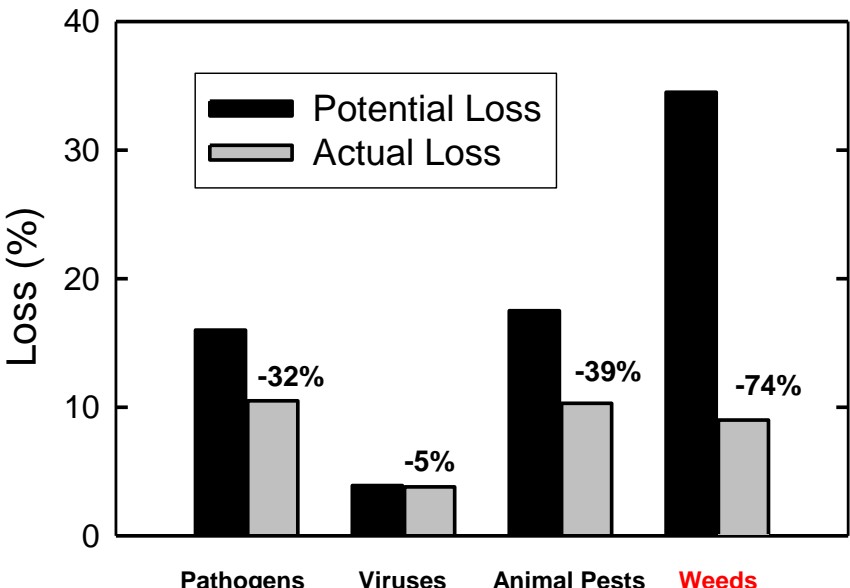

**Figure 1.** Potential and actual crop loss by pests. Differences in value reflect the ability to control the pests and reduce damage to crop production. Note that weed management provides the greatest reduction in weed-induced crop losses. Adapted from [16].

Maintaining the efficacy of such management efforts relative to rising $CO_2$ and climate change will be essential in promoting and sustaining crop production. However, there is an increasing number of reports indicating that rising $CO_2$ and projected changes in climate could shift weed demographics and increase weed competitiveness with negative consequences for agricultural productivity [17–20]. Moreover, there is evidence that climatic change and/or $CO_2$ may directly affect weed control, particularly chemical weed management, a predominant means for controlling weeds in developed countries [21–23].

Such studies, while preliminary, may necessitate a reevaluation of the current weed management paradigm. At present, that paradigm is centered around chemical weed management, especially in the United States and other countries where chemical control is extensively practiced.

It may be illustrative then to examine recent spatial and temporal trends on selection, evolutionary consequences, the current vulnerability of this paradigm, and the need for recognition and adoption of the consequences associated with rising $CO_2$ and climatic change for weed biology.

## 3. The Glyphosate Paradigm

Chemical weed management through the use of herbicides is among the most economical and widely used methods for weed control. In recent years, herbicide use has increased in other parts of the world, including China, India, and Sub-Saharan Africa, in part because of their potential to improve crop yields and save labor and energy, reducing overall pest management costs [24].

Arising from chemical research during World War II, the new and improved age of herbicides began. From 1945–1965, over 100 new substances were developed, tried, tested, and approved, including

two key chemicals, 2,4 -D (2,4 dichlorophenoxyacetic acid) and 2,3,4-T (2,4,5-trichlorophenoxyacetic acid). Their toxicity was such that even small amounts (1–2 kilos per hectare) were sufficient to control weeds [24]. The screening and identification of herbicides continued as part of overall research efforts to implement chemical weed management. However, beginning in the 1990s, the paradigm shifted.

This shift was related to the widespread use of a specific herbicide, glyphosate. As such, it is worthwhile to trace its history. Glyphosate was registered as an herbicide in 1974 and marketed under the trade name "Round-Up." Glyphosate acts as a dummy molecule in a biochemical pathway unique to plants and, in doing so, disrupts protein production [25]. However, it does not discriminate between weeds and crops, limiting its use in postemergence weed management.

Enter genetic engineering. Ongoing research efforts were successful in developing the first Round-Up-tolerant genetically modified (GMO) soybean line, released in the U.S. in 1996 as "Round-Up Ready." By 1997, 17% of all domestic soybean acres in the U.S. had transitioned to Round-Up Ready, with 68% having transitioned by 2001, 90% by 2010, and ~94% by 2019. Weed management was simple and highly effective relative to other methods. Such unequivocal adoption and use of soybean GMO spurred additional innovation, and Round-Up Ready corn and cotton swiftly followed [26].

With the success of this approach, after Round-Up Ready crops were introduced, a number of research articles were published reckoning that other management methods, such as crop rotations or the use of other herbicides, were simply not as effective [27,28]. As glyphosate dominated the herbicide market, other research efforts to identify and process new herbicides decreased.

However, as with many attempts to manage living organisms, chemical overuse can lead to rapid increases in resistance. Ten years after glyphosate was introduced, over 150 million pounds were being used in the United States alone, just in agricultural (not backyard) applications. In 2014, farmers sprayed enough glyphosate to apply ~1.0 kg/ha (0.8 pound/acre) on every hectare of U.S.-cultivated cropland and nearly 0.53 kg/ha (0.47 pounds/acre) on all cropland worldwide. As of 2020, that figure is probably closer to 300 million pounds [29].

In 1998, resistance to glyphosate was first observed in Italian ryegrass (*Lolium perenne* L. ssp. *multiflorum* (Lam.) Husnot) [30], followed by horseweed (*Conyza canadensis (L.) Cronquist*) in 2001 [31]. By 2020, almost 50 weed species had evolved resistance to glyphosate overall. At, present the Weed Science Society of America (WSSA) estimates over 500 unique cases (species x site of action) for herbicide resistance globally, with evolved resistance to 23 of the 26 known modes of action for 167 different herbicides. Overall, herbicide-resistant weeds have been observed in 94 different crops across 71 countries [32]. In the U.S., multiple-herbicide resistance and associated weeds, including water hemp (*Amaranthus tuberculatus* (Moq.) Sauer), velvet leaf (*Abutilon theophrasti)* Medik., morning glory (*Ipomea* spp.), and giant ragweed (*Ambrosia trifida* L.), are becoming endemic to much of the corn belt [33].

## 4. A Shifting Dynamic

If glyphosate dominance represented a stable paradigm, it was short-lived, but generated consequences. As a result of the widespread allure and efficacy of glyphosate, herbicide research was put on hold. However, as resistance reports multiplied, researchers quickly returned to the drawing board or wet lab, hoping to find new solutions, a new archetype for herbicidal control.

Such efforts are still ongoing. At present, older herbicides are being emphasized (Figure 2) and new transgenic crops resistant to these mixes being generated. Yet, such efforts are short term, in part because resistance already exists to these herbicides. Indeed, Round-Up (glyphosate) is still being sprayed to control weeds with resistance to other modes of action, but greater and greater concentrations are being used [33]. In addition, there are volatility and drift issues related to wider applications of older herbicides (e.g., dicamba), including damage to orchards and native trees [34,35]

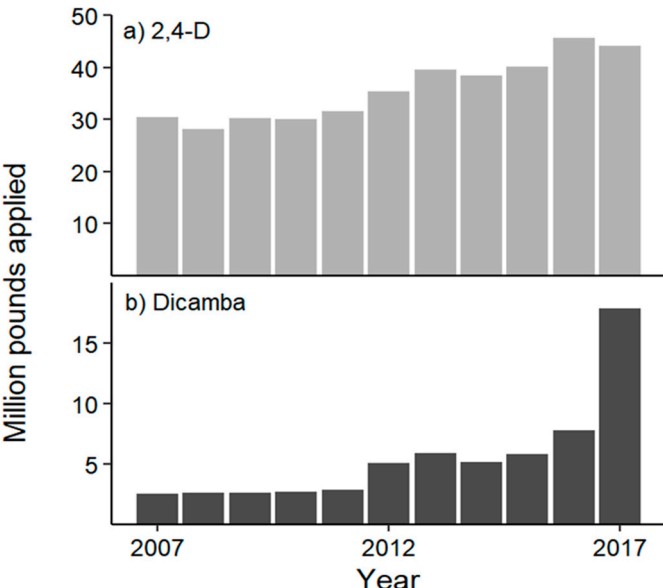

**Figure 2.** Change in application rates for two older mode of action (MOA) herbicides, 2,4-D and Dicamba. (Data are from the United States Geographical Survey (USGS) (https://water.usgs.gov/nawqa/pnsp/usage/maps/).

Some progress on new formulations is being made. For example, the first new mode of action (MOA) herbicide in over 35 years was recently announced by BASF ("Luximo") [36]. Other efforts are focusing on reducing drift in Dicamba, and new formulations and mixtures are being tried. Overall, chemical weed control for major crops is at a precarious stage, scrambling for new technology, a new paradigm.

## 5. CO$_2$, Climate: New Evolutionary Pressures

It is an optimistic axiom that we learn from our previous mistakes. In evaluating the history of glyphosate, one is reminded of another research effort in controlling pest (bacterial) threats, that of antibiotics, specifically penicillin. No sooner than the miraculous effects of penicillin became apparent to the general public, the antibiotic started to be overused. This triggered selective pressure for the emergence of penicillin-resistant strains, which, over a few years, spread across different countries. The discovery of each new generation of antibiotic quickly followed the same trend [37].

Not surprisingly, the basis for the decline in chemical effectiveness, irrespective of the biological organism to be controlled, has similar evolutionary patterns: The discovery of an effective control measure and the inherent belief that a "one-size-fits-all" paradigm has been established, followed by blanket application, overuse, and the rapid selection of resistance genotypes. In agronomic systems, blanket herbicide applications represent extraordinarily strong selective pressures, and the evolutionary potential of weeds is perhaps best illustrated by the rapid and widespread documentation of subsequent herbicide resistance [38].

If a known chemical begins to fail, it is tempting to return to the research strategy that brought success. For antibiotic pharmacological efforts, there are similar parallels to the herbicide industry, the low-hanging research fruit (soilborne antibiotics) have been utilized and greater biological efforts are underway. For example, extensive and rapid screening of synthetic compounds for antimicrobial properties or the use of genomic sequence information from different microbe strains to identify bacterial survival genes [39]. Clearly, parallel efforts (e.g., "gene-drive" technology [40] are underway for herbicide research [41].

Yet, there are new evolutionary pressures that must be considered in the context of herbicide research and efficacy, specifically the indirect effects of climate change and the direct effects of rising carbon dioxide. Such considerations are essential in maintaining chemical efficacy and future weed

management, as the overuse of herbicides and induced resistance have evidenced that weeds are capable of evolutionary change on ecological (generational) timescales [42–45].

Indirect climatic effects that influence evolutionary selection are already recognized. For example, changes in wind speed, rainfall, spray patterns, temperature effects on chemical persistence, etc., are inclusive to herbicide management. However, the projections of greater uncertainty in climate, including rapid temperature increases or precipitation extremes, will also impact all aspects of chemical weed management, from changes in crop systems to weed demography, weed-crop competition, the efficacy of spray applications, and physiological and phenological shifts in herbicide MOA, etc. However, to a large extent, these impacts and their evolutionary consequences are not well understood or characterized [21].

Similarly, the direct, evolutionary influence of rising $CO_2$ on plant biology, especially in the context of chemical weed management, is not always recognized within the research community. It is well established that photosynthesis by C3 plants (~90% of all plant species) is limited by the current concentration of $CO_2$, and that ongoing increases will stimulate photosynthesis and plant growth. However, the evolutionary roles of rising $CO_2$ on weed species selection, phenotypic changes in development and phenology, potential changes in crop/weed competition, and weed demographics are still being elucidated [18,43].

There are concerns directly related to herbicide efficacy and selection as well [21,44,46,47]. $CO_2$-induced changes in leaf thickness, root-shoot ratio, stomatal characteristics, etc., can alter the uptake, translocation, and dilution of herbicides, including glyphosate [21,44]. For famine weed, *Parthenium hysterophorus* L., a noxious invasive weed, $CO_2$-induced increases in biomass were a factor in recovery from glyphosate applications [48].

In addition, there are longer-term selection issues related to rising $CO_2$ levels and differential responses between weeds and crops. One of these is gene transfer, whereby changes in phenology and flowering may increase gene flow and alter herbicide resistance. This was observed between weedy red rice and Clearfield (an herbicide resistant rice line) as flowering times overlapped with increasing $CO_2$, with a subsequent increase in rice de-domestication and a greater number of weedy, herbicide-resistant hybrid progenies [49]. Another is $CO_2$ selection between herbicide-resistant and susceptible weed biotypes. For example, projected $CO_2$ levels increased the resistance of multiple-resistant strains of junglerice (*Echinochloa colona* L.) to cyhalofop-butyl [23].

There is a great deal more to be learned regarding rising $CO_2$, climate change, and weed biology. Unfortunately, many of these lessons are likely to be experienced in situ in the near future. However, there is sufficient evidence that increases in atmospheric $CO_2$ and a chaotic environment will impose strong selection pressures on weeds and that weeds will have the capacity to respond, in turn, with rapid adaptive evolution. This poses a substantial hurdle for chemical weed control. Indeed, it is not an exaggeration to suggest that these changes impose difficulties for chemical pest control in general.

## 6. A New Paradigm?

How can we seek a new weed management paradigm, one that recognizes current and future evolutionary pressures, a rapidly changing climate, and environmental consequences, yet is sustainable, effective, and economical?

It is important, in this context, not to claim a single idea or approach as a final answer to such a question. The ability to be dynamic and fluid in any management approach will be essential. Yet, there are some basic tenets that should be considered and included moving forward.

Let us recognize that historic pest demographics may not serve as a future standard. Rather, there is a clear and insistent need to provide ongoing assessments of new and emerging weeds. Evolutionary pressures by chemical overuse or climate change can significantly and suddenly alter specific weed threats (e.g., water hemp, giant ragweed). Hence, greater stress on early detection may be especially critical, as economic and environmental costs can increase exponentially with pest establishment. Following establishment, any existing or new threat requires appropriate evaluation and, if necessary,

weed control technology. Hence, the development of optimization systems for automated recognition of weeds and invasive plants is essential.

New technological advances in physical weed management show promise. Mechanical or robotic control of weeds facilitated by improved plant-weed recognition software and sophisticated global positioning system (GPS) is beginning to be recognized as a potential means to maintain weed control for specific crop systems (e.g., lettuce, [50]). Automatic weed removal technology provides a physical means for weed management, one that may be especially viable in specialty crops. Another mechanical innovation is the roller-crimper, a devise designed to cut cover crops as a means of weed suppression [51].

There are, at present, few economical alternatives to herbicides in large-acreage crops. Consequently chemical weed research will continue. It is tempting to find a single new chemical and/or derive a GMO crop that is suitably resistant as a means to further chemical pest management.

However, as we have experienced, such an approach has its own set of economic and biological risks. It costs millions of US dollars to develop a chemical product, perform rapid and widespread dissemination, and within a few years, the product may have developed evolutionary selection and massive resistance. At a time when sustainability is greatly desired as a means to control agronomic pests and maintain food security, particularly with an uncertain climate imposing additional selection pressures, sole reliance on a single MOA may be obsolete. Rather, diversity in any chemical endeavor may provide a more sustainable approach.

There are new avenues to begin diversification. Gene editing and genomic engineering may offer new insight into additional modes of action for herbicide development. New technologies that allow precise gene editing, such as CRISPER (clustered regularly interspaced short palindromic repeats), may be key to suppressing pest populations through "gene-drive" technology. CRISPR/Cas9 is efficient in inserting targeted mutations in both alleles of an individual, resulting in a conversion from the heterozygous to the homozygous condition with subsequent transmission of a specific gene to nearly all progeny. This technique has the potential to replace a given gene with a version designed by humans [52].

Another potential source remains biobased herbicides derived from natural plant products. Such products could represent allelopathic or novel microbial plant-pathogen chemistry. However, it is estimated that only a small fraction of microbial or plant biodiversity has been evaluated for herbicidal activity [41]. Here, there are also additional opportunities for genetic selection or engineering, e.g., the selection of crop plants with enhanced allelopathic (bioherbicide) capability. However, the role of climate and/or $CO_2$ on altering the chemical composition or availability of these products deserves additional consideration.

Whereas such efforts can provide additional tools in negating weed pressures, how these tools are managed constitutes another critical aspect of any future paradigm. The concept of diversification of management, of not putting all eggs in one basket, is key to sustainability. Such diversification is emblematic in integrated weed management (IWM), which assimilates different control tactics into a given management strategy. Such an approach can be simple (cleaning equipment, changing surfactants) or complex (decision support model systems, e.g., [53]). However, the central tenet of IWM is to alter selection pressures so as to prevent the dominance of a handful of highly adapted weed species. Such an evolutionarily aware approach, one that can utilize all available tools, is critical to minimize weed adaptation and spread [43].

## 7. Further Challenges

The brief synopsis of alternate approaches given here can be of potential use but is by no means complete. Additional efforts, especially in education and herbicide chemistry or new insight into weed biology and ecology, as well as climate uncertainty, are a vital aspect for next-generation scientists and land managers.

Perhaps the biggest challenge is one of human behavior. There is, understandably, an incentive for growers to use what worked before. Yet, dependence on a single management technique (e.g., herbicide application to GMO-resistant crops) will accelerate the evolution of herbicide resistance. Changing such behavior is a twostep process—derivation and testing of new approaches (as discussed earlier), as well as presentation, acceptance, and adoption of these approaches as effective at the field level. This process will be essential for any long-term sustainable approach to weed management, and given the onset of climate uncertainty, time is of the essence.

## 8. Final Thoughts

Weeds have been, and remain, the greatest constraint to global crop production. Their control and management will be essential if food security is to be maintained with projected increases in human population for the current century.

There is no question that chemical application has been a cornerstone of effective weed control within developed countries for decades. The combination of chemical use with GMO-resistant crops has been, until recently, the epitome of the weed management paradigm.

However, as is increasingly clear, overreliance on a single methodology and subsequent rapid increases in herbicide resistance belie the long-term sustainability of such an approach. The ability of weeds to undergo rapid genetic change and to develop resistance to multiple mechanisms of herbicide action stresses the need for a reassessment of past practices. There are a number of seminal reviews [41,54–58] that recognize the current, critical need to reassess and transform weed control technology, particularly herbicide efficacy, to cope with spatial and tempoThiral changes in weed evolution and adaptation.

Given this need, it is a seminal time to simultaneously diagnose the challenges of rising levels of $CO_2$, and climate uncertainty for all aspects of weed management. It is a seminal time because any reassessment and any need to formulate a new paradigm must recognize these environmental parameters as significant in their own ability to alter weed evolution and herbicide efficacy. As such, diagnosing the challenges of rising levels of $CO_2$, and climate uncertainty is essential to any long-term sustainable weed management effort.

**Funding:** This research received no external funding.

**Acknowledgments:** The author thanks Julie Wolf, USDA-ARS for her assistance in the preparation of this manuscript.

**Conflicts of Interest:** The author declares no conflict of interest.

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
