# Peer review of "Climate Change and the Herbicide Paradigm: Visiting the Future"

_agronomy, doi:10.3390/agronomy10121953_

Round 1

Reviewer 1 Report

This is a review/opinion manuscript about herbicide and climate change. I believe the review is a great contribution to weed science discipline. I have made some notes in the attached document, please check it. Overall I think the reading is flowing well. Please check my comments. 

Author Response

Response to Reviewer No. 1

Please Note: Comments were made onto a pdf of the manuscript.  I have gone through and addressed them as follows:

  1. "gmo" on line 8 of the abstract has been rewritten as requested. 
  2. An additional sentence elucidating the difference between potential and actual pest reduction has been added to the figure legend for figure 1.
  3. Lines 81-86 (old manuscript).  This has been rewritten to include a number of the reviewers comments. Lines 113-123 in new manuscript. 
  4. Lines 98-101.  Details, including scientific names have been added.
  5. Line 113 old manuscript.  An additional reference for dicamba has been added. 
  6. Line 176 (old manuscript), common name for Echinochloa colona has been added.
  7. Line 194 (old manuiscript), scientific name added.
  8. Line 201 (old manuscript), full name of GPS is given.
  9. Line 207, 243, 253,  (old manuscript), gmo is now capitalized.
  10. Note that additional references have been added, and renumbered in new manuscript.

Reviewer 2 Report

This is an interesting paper written by one of the experts in the field of climate change and weed management. There are several interesting thoughts and ideas that needs to be elaborated and explored more thoroughly. For instance, in the part discussing the glyphosate paradigm, more explanations are needed to understand the context of the paradigm and how it is associated with climate change.

There is a need to show more innovation, maybe discussing new application technologies, formulations or the use of nanoparticles? Something more than one new MOA that was announced by BASF.

I think that more discussion about the correlation between plant phenology and physiology explaining the effect of climate change on herbicide efficacy would benefit this perspective. Herbicide translocation and metabolism may differ under various phenological stages and environmental conditions, this is probably the explanation for reduced efficacy.

Not all readers are familiar with the literature related to chemical weed management history. In my opinion, elaborating in this matter would contribute to the understanding of the process and shifts in paradigm. For instance, the shift from the use in PSII inhibitors to ALS inhibitors made a major difference in chemical weed management.

The effect of CO2 is widely discussed, however, the other main and maybe more influencing variable of temperature shifts, is poorly discussed. A section on the effect of temperature on herbicide response should be added.

Across the text, it seems that there are several places were the font is not the same, should be corrected.

20-21: please add references.

33-37: weed damage should be highlighted instead of damages by other pests, I suggest using ref 22 here instead.

44-46: if examples are given from research papers they should be further discussed.

46-47: this sentence is confusing, did you mean to show that these damages do not occur when weed control efforts are in place?

67: other than where?

77-80: please add references.

82: what does GMO stands for? Capital letters (GMO)

82-83: please add proper references.

85: GMO.

92: should not be italics.

97: please insert a coma after "in 1998".

97: it was actually reported prior to that by Pratley et al. "Pratley, J., P. Baines, P. Eberbach, M. Incerti, and J. Broster. 1996. Glyphosate resistance in annual ryegrass. Page 126 in J. Virgona and D. Michalk, eds. Proceedings of the 11th Annual Conference of the Grasslands Society of New South Wales. Wagga Wagga, Australia: The Grasslands Society of NSW"

98-99: it may be better to refer to the international herbicide-resistant weed database that show 326 cases reported until 2019.

108-109: 2,4-D and dicamba are not very good examples for this trend as they been used consequently since the 1940 (more or less). Herbicides such as Boxer (Prosulfocarb) seems better.

110: there are numerus different formulation and trade names today, it would be better to use the active ingredient instead of product name. There is actually a product called Roundup in the US that contains only auxinic inhibitors.

114-115: what is the MOA? Which process? Any reference for this? Dayan have published a very nice paper in this subject "Current Status and Future Prospects in Herbicide Discovery" showing all new MOA.

127: 2,4-D and not 2,3-D.

130-133: discussing herbicide selectivity should be followed by the introduction of genetic components affecting this selectivity (Jasieniuk M, Bruˆle´-Babel AL, Morrison IN (1996) The evolution and genetics of herbicide resistance in weeds. Weed Sci 44:176–193).

143: ref No. 37 discuss insects, are there better references for this?

147: it is not accurate to say that "herbicide induce resistance", however, overuse or misuse of them might do that.

148: my suggestion is to replace the review papers with research papers showing transgenerational changes in weed populations caused due to weed management practices or environmental changes, such as:

Tehranchian, P.; Norsworthy, J.K.; Powles, S.; Bararpour, M.T.; Bagavathiannan, M.V.; Barber, T.; Scott, R.C. Recurrent sublethal-dose selection for reduced susceptibility of palmer amaranth (Amaranthus palmeri) to dicamba. Weed Sci. 2017, 65, 206–212.

Matzrafi, M., Osipitan, O., Ohadi, S., & Mesgaran, M. (2020). Under pressure: Maternal effects promote drought tolerance in progeny seed of Palmer amaranth (Amaranthus palmeri). Weed Science, 1-29. doi:10.1017/wsc.2020.75.

Busi, R.; Gaines, T.A.; Walsh, M.J.; Powles, S.B. Understanding the potential for resistance evolution to the new herbicide pyroxasulfone: Field selection at high doses versus recurrent selection at low doses. Weed Res. 2012, 52, 489–499.

149-151: please add references.

151-155: this sentence is too long, please consider rephrasing.

159-161: please add references, preferably a review paper.

165-166: a good example for the influence of both temperatures and CO2 levels on glyphosate activity can be found in "Matzrafi et al. Increased temperatures and elevated CO2 levels reduce the sensitivity of Conyza canadensis and Chenopodium album to glyphosate. Sci Rep 9, 2228 (2019). https://doi.org/10.1038/s41598-019-38729-x.

167-168: Bajwa et al reported on this earlier, Bajwa AA, Wang H, Chauhan BS, Adkins SW. Effect of elevated carbon dioxide concentration on growth, productivity and glyphosate response of parthenium weed (Parthenium hysterophorus L.). Pest Manag Sci. 2019 Nov;75(11):2934-2941. doi: 10.1002/ps.5403.

171: my suggestion, herbicide response instead of "herbicide resistance".

175: increase the resistance and not "resistant".

184-235: this section seems too general. There are many non-chemical weed management approaches, however, using them instead of herbicides can't be the whole solution. If climate change will affect herbicide penetration, translocation and metabolism, how can we cope with that? Is there any chemical or biochemical way the bypass or reduce the environmental effect?

194: palmer is a good example, however, author should elaborate to show why and how. Also, more weeds such as Ambrosia and Echinochloa species should also be referred to.

207: GMO instead of gmo.

221: I suggest a different reference for the CRISPR method, Tang, X., Lowder, L., Zhang, T. et al. A CRISPR–Cpf1 system for efficient genome editing and transcriptional repression in plants. Nature Plants 3, 17018 (2017). https://doi.org/10.1038/nplants.2017.18.

223-225: this is a fascinating issue, I think the author should elaborate more on this topic and maybe also discuss the influence of climate change on these compounds and microorganisms.

243: GMO instead of gmo.

241-247: it would have been great to get more discussion on the social and human behavior aspect

253: GMO instead of gmo.

Author Response

I have done my best to respond to Reviewer No. 2's concerns.  Please note however that often these concerns are related to differences in how to approach or provide an opinion on a given topic, and feel that in a perspective piece, some leeway should be given to the author.  

Note that the original review comments are italicized.

This is an interesting paper written by one of the experts in the field of climate change and weed management. There are several interesting thoughts and ideas that needs to be elaborated and explored more thoroughly. For instance, in the part discussing the glyphosate paradigm, more explanations are needed to understand the context of the paradigm and how it is associated with climate change.

There is a need to show more innovation, maybe discussing new application technologies, formulations or the use of nanoparticles? Something more than one new MOA that was announced by BASF.

I think that more discussion about the correlation between plant phenology and physiology explaining the effect of climate change on herbicide efficacy would benefit this perspective. Herbicide translocation and metabolism may differ under various phenological stages and environmental conditions, this is probably the explanation for reduced efficacy.

Not all readers are familiar with the literature related to chemical weed management history. In my opinion, elaborating in this matter would contribute to the understanding of the process and shifts in paradigm. For instance, the shift from the use in PSII inhibitors to ALS inhibitors made a major difference in chemical weed management.

Response: All of these are interesting and important points to make; however, the article is a perspective, an opinion.  One can argue for example that a shift from PSII to ALS inhibitors made a difference in chemical weed management, but at present it matters very little as resistance to both is roughly equal. The point of the piece (simplified) is that weed science needs to go back to the drawing board, because of the rapid development and spread of chemical resistance--and since it does, maybe it should consider climate change in future approaches. 

Overall, the piece reflects my own viewpoint and thoughts.  There are, and are referred to in the article, other publications that encompass these points made by the reviewer.  Unfortunately, as with any opinion piece, it is difficult to address all points of view.  

The effect of CO2 is widely discussed, however, the other main and maybe more influencing variable of temperature shifts, is poorly discussed. A section on the effect of temperature on herbicide response should be added.

Response: One could also argue of a greater need to discuss physical disruptions in addition to temperature (e.g. wind, extreme precipitation events).  However, please note that there already is a paragraph (lines 179-186 in the revised manuscript that deals with temperature and other physical phenomenon associated with climate change. These are well-known and written about--less recognized are the CO2 aspects.  

Across the text, it seems that there are several places were the font is not the same, should be corrected. 

Response: Noted, and the font should now be Calibri 11 throughout. 

20-21: please add references.

Response: A reference from the UN has been added.

33-37: weed damage should be highlighted instead of damages by other pests, I suggest using ref 22 here instead.

Response: Respectfully disagree, in an Introduction it is a better narrative to go from a general (pest issues) to a specific (weeds) topic. This segues in turn to the efficacy of different pest management (figure 1) and the importance of herbicides. 

44-46: if examples are given from research papers they should be further discussed.

Response: The papers mentioned simply refer to other negative consequences of weeds in agronomic settings, there is no need to provide additional details.  

46-47: this sentence is confusing, did you mean to show that these damages do not occur when weed control efforts are in place?

Response: An additional sentence has been added to the figure caption to provide context.  Hopefully it is now clear to the reviewer.

67: other than where?

Response. Unclear--is the reviewer asking for a quantitative analysis for all of the places where ag is practiced where herbicides are not used?  

77-80: please add references.

Response: The Duke 2018 reference on the history of glyphosate has been added.

82: what does GMO stands for? Capital letters (GMO)

Response: GMO as per reviewer no. 1, has been amended throughout the manuscript.

82-83: please add proper references.

Response: As per the first reviewer, references and section have been rewritten for clarity and relevance. 

92: should not be italics.

Response: Italics have been removed. 

97: please insert a coma after "in 1998".

Response: comma inserted.

97: it was actually reported prior to that by Pratley et al. "Pratley, J., P. Baines, P. Eberbach, M. Incerti, and J. Broster. 1996. Glyphosate resistance in annual ryegrass. Page 126 in J. Virgona and D. Michalk, eds. Proceedings of the 11th Annual Conference of the Grasslands Society of New South Wales. Wagga Wagga, Australia: The Grasslands Society of NSW"

Response:  Yes, Proceedings and other notifications of resistance usually occur prior to peer-reviewed publications; however, I feel that the reference provided is widely recognized for being the first to report resistance to glyphosate. 

98-99: it may be better to refer to the international herbicide-resistant weed database that show 326 cases reported until 2019.

Response: This section has been rewritten and updated.

108-109: 2,4-D and dicamba are not very good examples for this trend as they been used consequently since the 1940 (more or less). Herbicides such as Boxer (Prosulfocarb) seems better.

Response: Respectfully disagree, 2-4 D and Dicamba are some of the earliest herbicides, and exemplary (in my opinion) of the "back to the drawing board" response of much of the herbicide industry when glyphosate resistance occurred.

110: there are numerus different formulation and trade names today, it would be better to use the active ingredient instead of product name. There is actually a product called Roundup in the US that contains only auxinic inhibitors. 

Response: The point here is to illustrate that more and more glyphosate is being used.  However, I have added "glyphosate" to the sentence for clarity.

114-115: what is the MOA? Which process? Any reference for this? Dayan have published a very nice paper in this subject "Current Status and Future Prospects in Herbicide Discovery" showing all new MOA.

Response: The Dayan paper is appreciated, it encapsulates much of the current perspective actually, i.e., glyphosate paradigm is no longer, we are back to basics, and here are some possibilities.  I would only  stress that the Dayan paper shows potential MOA, none that have been approved--Luximo, to the best of my knowledge is first.  Modes of action have been abbreviated now to MOA throughout the ms.   I have also suggested the Dayan paper as another review paper at the end of the current manuscript.

127: 2,4-D and not 2,3-D.

Response: corrected. 

130-133: discussing herbicide selectivity should be followed by the introduction of genetic components affecting this selectivity (Jasieniuk M, Bruˆle´-Babel AL, Morrison IN (1996) The evolution and genetics of herbicide resistance in weeds. Weed Sci 44:176–193).

Response: This section is not about herbicide selectivity per se, simply drawing parallels with antibiotic efforts within the medical profession.  

143: ref No. 37 discuss insects, are there better references for this?

Response:  Reference is used because insecticide development has made great strides in gene-driven approaches---it is appropriate to use here. 

147: it is not accurate to say that "herbicide induce resistance", however, overuse or misuse of them might do that.

Response: Good point. Line 175 in revised manuscript has been rewritten for clarity. 

148: my suggestion is to replace the review papers with research papers showing transgenerational changes in weed populations caused due to weed management practices or environmental changes, such as:

Tehranchian, P.; Norsworthy, J.K.; Powles, S.; Bararpour, M.T.; Bagavathiannan, M.V.; Barber, T.; Scott, R.C. Recurrent sublethal-dose selection for reduced susceptibility of palmer amaranth (Amaranthus palmeri) to dicamba. Weed Sci. 2017, 65, 206–212.

Matzrafi, M., Osipitan, O., Ohadi, S., & Mesgaran, M. (2020). Under pressure: Maternal effects promote drought tolerance in progeny seed of Palmer amaranth (Amaranthus palmeri). Weed Science, 1-29. doi:10.1017/wsc.2020.75.

Busi, R.; Gaines, T.A.; Walsh, M.J.; Powles, S.B. Understanding the potential for resistance evolution to the new herbicide pyroxasulfone: Field selection at high doses versus recurrent selection at low doses. Weed Res. 2012, 52, 489–499.

Response:  I appreciate these suggestions, but believe the focus should be on evolution in response to selection pressures (i.e. herbicides) per se.  The references currently being used reflect that choice. 

149-151: please add references.

Response: I do not think that references that assert that wind speed, rainfall, temperature (i.e. climate) affect herbicide application are necessary. 

151-155: this sentence is too long, please consider rephrasing.

Response: Sentence has been rewritten and shortened.

159-161: please add references, preferably a review paper.

Response:  If photosynthetic response to CO2 is in question, then a review, or several reviews would be warranted; however, it is somewhat akin to saying there are several hundred papers showing plants respond to light.  Is there really a need for referencing review papers? However, I have rewritten the sentence accordingly (lines 189-190) in revision.

165-166: a good example for the influence of both temperatures and CO2 levels on glyphosate activity can be found in "Matzrafi et al. Increased temperatures and elevated CO2 levels reduce the sensitivity of Conyza canadensis and Chenopodium album to glyphosate. Sci Rep 9, 2228 (2019). https://doi.org/10.1038/s41598-019-38729-x.

Response:  The Matzrafi reference has been added. 

167-168: Bajwa et al reported on this earlier, Bajwa AA, Wang H, Chauhan BS, Adkins SW. Effect of elevated carbon dioxide concentration on growth, productivity and glyphosate response of parthenium weed (Parthenium hysterophorus L.). Pest Manag Sci. 2019 Nov;75(11):2934-2941. doi: 10.1002/ps.5403.

Response: Fair point, the Bajwa et al. reference will replace the Cowie et al. reference.

Round 2

Reviewer 2 Report

Accept